# A Pathogenic Variant Reclassified to the Pseudogene *PMS2P1* in a Patient with Suspected Hereditary Cancer

**DOI:** 10.3390/ijms24021398

**Published:** 2023-01-11

**Authors:** Veronica Fragoso-Ontiveros, Marcela Angelica De la Fuente-Hernandez, Vincent Gonzalez-Osnaya, Mario Gamez-Rosales, Maria Delia Perez-Montiel, David Isla-Ortiz, David Francisco Cantu-De Leon, Rosa Maria Alvarez-Gomez

**Affiliations:** 1Hereditary Cancer Clinic, National Cancer Institute, San Fernando 22, Sección XVI, Tlalpan, Mexico City 14070, Mexico; 2Pathology Direction, National Cancer Institute, Mexico City 14080, Mexico; 3Gineco-Oncology Department, National Cancer Institute, Mexico City 14080, Mexico; 4Research Direction, National Cancer Institute, Mexico City 14080, Mexico

**Keywords:** *PMS2*, pseudogene, pathogenic variant, Lynch Syndrome, reclassification, hereditary cancer

## Abstract

The *PMS2* gene is involved in DNA repair by the mismatch repair pathway. Deficiencies in this mechanism have been associated with Lynch Syndrome (LS), which is characterized by a high risk for colorectal, endometrial, ovarian, breast, and other cancers. Germinal pathogenic variants of *PMS2* are associated with up to 5% of all cases of LS. The prevalence is overestimated for the existence of multiple homologous pseudogenes. We report the case of a 44-year-old woman diagnosed with breast cancer at 34 years without a relevant cancer family history. The presence of pathogenic variant NM_000535.7:c.1A > T, (p.Met1Leu) in *PMS2* was determined by next-generation sequencing analysis with a panel of 322 cancer-associated genes and confirmed by capillary sequencing in the patient. The variant was determined in six family members (brothers, sisters, and a son) and seven non-cancerous unrelated individuals. Analysis of the amplified region showed high homology of *PMS2* with five of its pseudogenes. We determined that the variant is associated with the *PMS2P1* pseudogene following sequence alignment analysis. We propose considering the variant c.1A > T, (p.Met1Leu) in *PMS2* for reclassification as not hereditary cancer-related, given the impact on the diagnosis and treatment of cancer patients and families carrying this variant.

## 1. Introduction

Lynch syndrome (LS), or Hereditary Non-Polyposis Colorectal Cancer (HNPCC), is an autosomal dominant inherited cancer syndrome. LS is associated with an increased risk of colorectal, endometrial, ovarian, and breast cancer [1,2,3,4]. Germinal pathogenic variants in the DNA mismatch repair (MMR) pathway are involved in LS. The central genes in LS are *MLH1*, *MSH2*, *MSH6*, *PMS2*, and *EPCAM* [5,6]. Since the genetic predisposition for LS is related to pathogenic variants in one of the four MMR genes, most LS cancers show an MMR deficiency, microsatellite instability, and activation of the immune response system. Hence, LS cancer patients may be optimal candidates for immune checkpoint-based therapies [7]. The *PMS2* (PMS1 Homolog 2) gene is located on chromosome 7 with an approximate length of 38,000 base pairs contained in 15 exons. The *PMS2* protein product is 862 amino acids long [8]. The *PMS2* protein is an MMR system component, forming part of the MutLα heterodimer, through the interaction of the C-terminal domain of its gene with MLH1 [9]. The MutLα heterodimer possesses an endonucleolytic activity that is necessary for the removal of the mismatched DNA (https://www.ncbi.nlm.nih.gov/gene/5395 accessed on 3 January 2023). *PMS2* variants have been associated with breast and ovarian cancer in LS, with a frequency of up to 47.6% [10]. Cancer risk assessment for *PMS2* variants is quite complex due to their low penetrance, and the sequencing analysis is challenging because of the presence of multiple pseudogenes [11]. Pseudogenes are defined as incomplete copies of genes that code for proteins. However, the pseudogenes do not code for functional proteins due to changes in the sequence in addition to having lost essential regulatory elements for translation [12]. *PMS2* has fourteen pseudogenes that share high homology with the 5’ end, which spans exons 1 through 5, while a fifteenth pseudogene, *PMS2CL*, shares high homology with exon 9 and exons 11–15 [13,14]. This huge number of pseudogenes complicates the precision of the molecular analysis of the *PMS2* gene in the search for variants associated with the development of LS and other types of cancer [15].

## 2. Case Presentation

A 44-year-old Mexican woman was diagnosed at 34 with left unilateral breast cancer, IIIA clinical stage (T3N1M0). The tumor was an infiltrating ductal carcinoma, with Scarff-Bloom-Richardson (SBR) 8, hormone receptor-positive, HER2-negative, and Ki67 40% characteristics. The patient received neoadjuvant treatment with chemotherapy (FAC regimen), followed by a modified left radical mastectomy. Subsequently, radiotherapy was prescribed at a total dose of 50Gy, in 25 fractions. She underwent hormone therapy with tamoxifen for five years. At the age of 40 years, the patient presented with abnormal uterine bleeding, so a hysterectomy was performed with bilateral salpingectomy and left oophorectomy, finding an endometrial polyp. The patient received a cancer risk assessment at the Hereditary Cancer Clinic of the National Cancer Institute (INCan). The pedigree identified two female cousins, on the paternal side, with a history of breast cancer. The first was diagnosed at 58 years and is still alive; the second was at 45 years and died at 48. On the maternal side, she referred to a second cousin with unspecified cancer at 40, who died at 42 years (Figure 1). Due to the early age presentation of cancer and breast cancer family history, the patient was considered a high-risk patient for hereditary cancer, and molecular testing for the multigene panel was offered. Following the informed consent process, genomic DNA was obtained from peripheral blood and analyzed by Next-Generation Sequencing (NGS) through the Illumina-Nimblegen commercial platform (Illumina, San Diego, CA, USA), using panels of 322 cancer-associated genes (Appendix A). The libraries were sequenced using paired-end, 100-cycle chemistry on the Illumina HiSeq 2000 platform (Illumina, San Diego, CA, USA). Germline mutations and indels were called using Platypus [16] to create a VCF file (Variant Call Format 4.0), and were then annotated with ANNOVAR version 06-01-2017 [17] with multiple databases, including the Single Nucleotide Polymorphism Database version 150 (dbSNP), 1000 Genomes, and HapMap, COSMIC version 81, ClinVar version 06-03-2018, Intervar version 01-18-2018, and RefSeq version 91. Identification of potentially deleterious effects on protein function was performed using in silico prediction algorithms: SIFT [18,19] and PolyPhen-2 [20].

## 3. Results and Discussion

The heterozygous germline variant in *PMS2* NM_000535.7:c.1A > T, (p.Met1Leu) (ClinVar accession CVC000142777.15; rs587779333) was reported and classified as pathogenic in ClinVar (http://www.ncbi.nlm.nih.gov/clinvar/ accessed on 20 December 2022). The variant was corroborated by capillary sequencing from PCR amplification of the region of interest using specific primers. The sequence was compared with the *PMS2* reference mentioned above (Figure 2a).

Based on the personal and family history of cancer, together with the identification of the pathogenic variant in *PMS2*, the patient was considered to have LS. The molecular testing set was extended to six family members (five siblings and one son of the proband) to determine if they carried the variant c.1A > T, (p.Met1Leu) in *PMS2*. Interestingly, this variant was detected in all six family members examined, among which, a chromatogram of a single family member is exemplified as a representative (Figure 2b). In detail, the molecular testing results of the proband and the family members are shown (Table 1 and Appendix A). Because each family member tested harbored the variant in a disease-free status, we examined the presence of the variant in seven healthy volunteers unrelated to the study family; that is, not diagnosed with cancer. The presence of the variant was found in 7/7 of the samples evaluated. A representative chromatogram of an unrelated individual is shown (Figure 2c) and all individuals are detailed (Table 1 and Appendix A).

The variant c.1A > T, (p.Met1Leu) in *PMS2* affects the methionine involved in translation initiation of this gene. Functional assays for this variant have not been reported; however, the nucleotide change is predicted to generate a non-functional truncated protein. The immunohistochemistry (IHC) analysis correlates 75% of the loss of *PMS2* expression with a pathogenic variant in the same gene [15]. IHC showed the nuclear expression of the *PMS2* protein (Appendix A). This variant has been informed only in an individual with suspected LS [21] and another individual with a second pathogenic variant of *PMS2* and a personal history consistent with MMR deficiency syndrome [22]. The rs587779333 variant has reported the alleles T > A; T > C; and T > G. The reference allele in the Latino population is 100% and 0% for the alternate alleles T and G, respectively, according to the Allele Frequency Aggregator (ALFA) database (https://www.ncbi.nlm.nih.gov/snp/docs/gsr/alfa/ accessed on 20 December 2022). In the gnomAD database, the frequency is the same, but there are no data for the alternative “T” allele. Notably, the Latino population is underrepresented compared with the European, African, and African American populations in both databases (https://www.ncbi.nlm.nih.gov/snp/rs587779333 accessed on 1 August 2022; National Center for Biotechnology Information).

We found a frequency of 100% of the variant in the analyzed individuals, relatives and non-relatives alike, within the study sample. In accordance with these results, we raise the possibility that the c.1A > T, (p.Met1Leu) variant in *PMS2* reported as pathogenic is, in fact, a change of sequence in one of the *PMS2* pseudogenes. Multiple sequence alignment analysis (MUSCLE, https://www.ebi.ac.uk/Tools/msa/muscle/ accessed on 8 July 2022) showed homology to 5 pseudogenes: *PMS2P1, P2, P4, P5*, and *P7* (Table 2, Figure 3a). Subsequently, we determined whether the PCR-amplified region (g. −21 to g. 175, *PMS2* NG_008466) was common to these five pseudogenes using the Basic Local Alignment Search Tool (BLAST, https://blast.ncbi.nlm.nih.gov/Blast.cgi?PAGE_TYPE=BlastSearch accessed on 8 July 2022) alignment algorithm. We found that *PMS2P1* is the only pseudogene to share the region of interest with *PMS2*, with 89% identity. In the region of *PMS2* corresponding to the ATG at the start of transcription, *PMS2P1* contains the TTG nucleotides (Figure 3b). Finally, we estimated the degree of complementarity of the previously used primers with the sequence of the *PMS2P1* pseudogene. Results demonstrated 100% and 80% of complementarity with the forward and reverse primers, respectively, allowing simultaneous amplification of the exon one sequence, which corresponds to the homologous region between *PMS2* and *PMS2P1* (Figure 3c).

Misclassification of variants in genes with many pseudogenes of the homologous sequence is plausible due to the difficulty and limitations of current sequencing technologies to discriminate a genomic region from a pseudogene. Reclassification of variants occurs in genes with a high presence of pseudogenes. Still, it has also been necessary for well-characterized genes such as *BRCA1* and *BRCA2*, which have presented ethical and practical challenges related to the clinical management of patients carrying these variants [23]. The *PMS2* variant NM_000535.5:c.2182_2184delACTinsG is estimated to be found in 2.5% of the African population. It is classified as pathogenic in ClinVar and has been proposed for reclassification because a long PCR analysis determined that it is a variant in the *PMS2CL* pseudogene [24].

## 4. Conclusions

The biological significance of genetic variants and their implications for cancer risk render an essential role in medical diagnosis and treatment. In circumstances where a genetic variant is a matter of controversy or has doubt regarding its clinical consequences, it will be crucial to raise the concern in the appropriate outcome report [25]. In the case we presented, the patient had a history of a partial oophorectomy, so during the initial post-test genetic counseling and in contemplating a molecular result related to LS, the possibility of performing a contralateral-oophorectomy was evaluated according to management guidelines [26]. In addition, the impact influences treatment decisions, with the current evidence of the use of immune checkpoint therapy for tumors associated with LS [7]. For these reasons, we propose further investigating the reported variant c.1A > T, (p.Met1Leu) in *PMS2* and evaluating its reclassification to allow appropriate clinical and surgical management for the patient and family members.

## Figures and Tables

**Figure 1 ijms-24-01398-f001:**
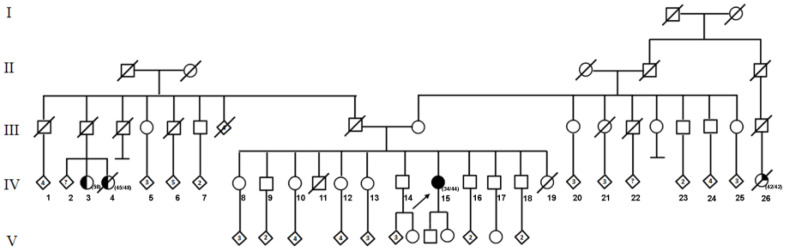
Pedigree of the family of patient suspected of hereditary cancer. The proband IV-15 is indicated by the arrow. The individuals IV-3 and IV-4 were diagnosed with unilateral breast cancer and the individual IV-26 with an unspecified cancer. The age of diagnosis and the current age of affected individuals is given in brackets.

**Figure 2 ijms-24-01398-f002:**
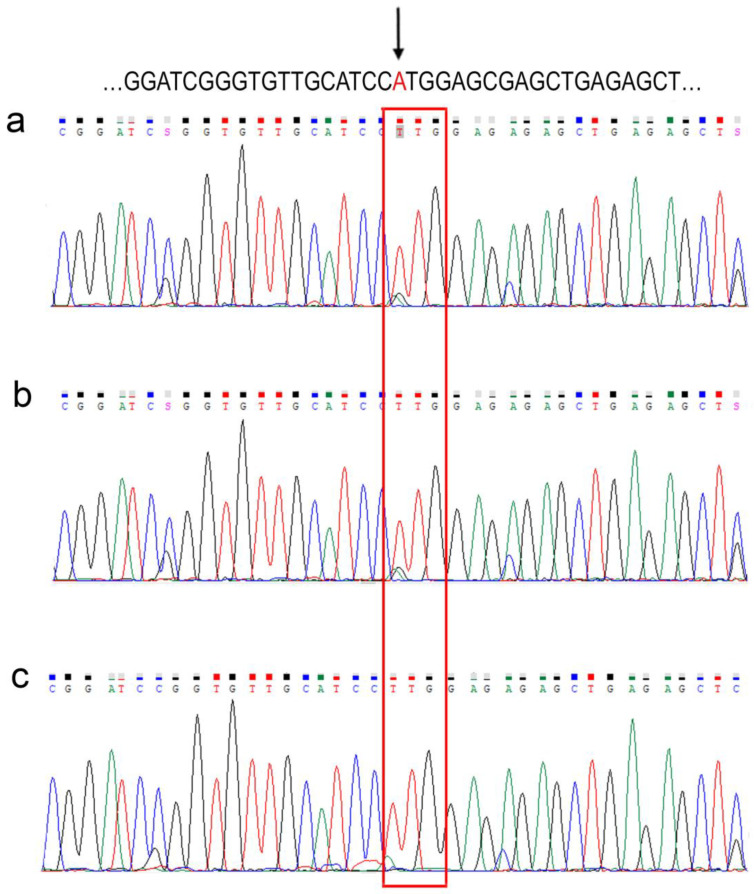
Molecular diagnosis of the variant c.1A > T, (p.Met1Leu) in *PMS2.* Up sequence reference NG_008466.1. Sequencing chromatograms for (**a**) index patient; (**b**) family extension; and (**c**) healthy volunteer. Arrow indicates c.1A > T, (p.Met1Leu) in *PMS2*; the red rectangle marks the change region c.1A > T, (p.Met1Leu).

**Figure 3 ijms-24-01398-f003:**
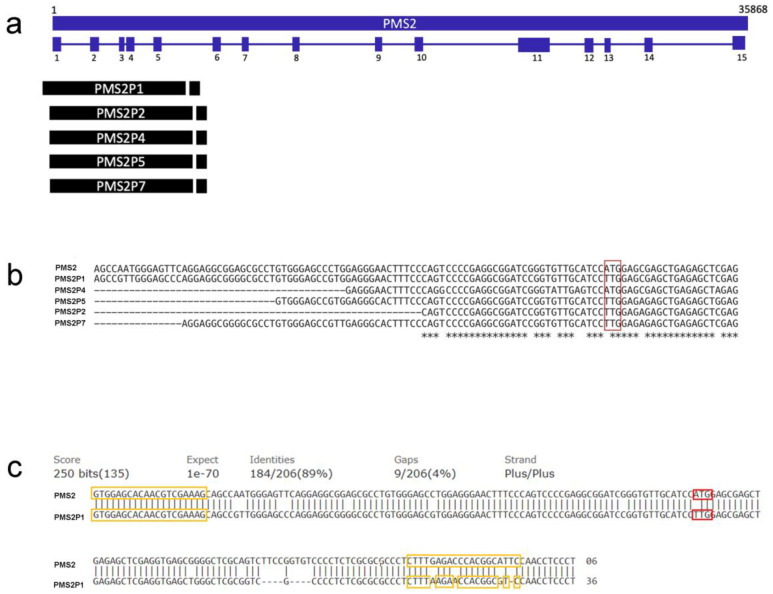
A schematic diagram of the genomic structure of *PMS2* and the pseudogenes. (**a**) The blue boxes show the 15 exons of the *PMS2* gene, whose position is representative. The black boxes show the pseudogenes that match exon 1 and exon 5. (**b**) Multiple sequence alignment of exon 1; red box denotes c.1A > T, (p.Met1Leu) in *PMS2*; asterisk (⁎) denotes conserved nucleotide in all sequences. (**c**) Alignment of the PCR amplicon with the reference sequence of the pseudogene *PMS2P1*. The yellow boxes indicate the complementarity of the forward and reverse primers. The red box indicates the site of change c.1A > T, (p. Met1Leu).

**Table 1 ijms-24-01398-t001:** Prevalence in family tracts and unrelated healthy individuals of the variant c.1A > T, (p.Met1Leu) in *PMS2*.

ID	Gender	Age (Years)	Diagnosis	Relationship	Sample	PMS2 c.1A > T, (p.Met1Leu)
1	F	44	Breast cancer	Proband	PB	positive
2	M	19	healthy	son	PB	positive
3	M	37	healthy	brother	PB	positive
4	M	40	healthy	brother	PB	positive
5	M	42	healthy	brother	PB	positive
6	F	49	healthy	sister	PB	positive
7	F	52	healthy	sister	PB	positive
C1	M	29	healthy	UC	PB	positive
C2	M	30	healthy	UC	PB	positive
C3	M	36	healthy	UC	PB	positive
C4	M	37	healthy	UC	PB	positive
C5	F	35	healthy	UC	PB	positive
C6	F	47	healthy	UC	PB	positive
C7	F	55	healthy	UC	PB	positive

F = female; M = male; C = healthy controls; UC = unrelated control; PB = peripheral blood.

**Table 2 ijms-24-01398-t002:** *PMS2* pseudogenes. NCBI reference numbers and sequences used for alignment and analysis of exon 1 overlap regions.

Gene/Pseudogene	Description	NCBI Reference Sequence	Match Exon 1
*PMS2*	Homo sapiens PMS1 homolog 2, mismatch repair system component (PMS2), RefSeqGene (LRG_161) on chromosome 7	NG_008466.1	Reference
*PMS2P1*	PMS1 homolog 2, mismatch repair system component pseudogene 1 [Homo sapiens]	NC_000007.14:c100336307-100320640	Yes
*PMS2P2*	PMS1 homolog 2, mismatch repair system component pseudogene 2 [Homo sapiens]	NC_000007.14:c75358997-75343937	Yes
*PMS2P3*	PMS1 homolog 2, mismatch repair system component pseudogene 3 [Homo sapiens]	NC_000007.14:c75528123-75507747	No
*PMS2P4*	PMS1 homolog 2, mismatch repair system component pseudogene 4 [Homo sapiens]	NC_000007.14:c67302442-67276131	Yes
*PMS2P5*	PMS1 homolog 2, mismatch repair system component pseudogene 5 [Homo sapiens]	NC_000007.14:74890768-74921138	Yes
*PMS2P6*	PMS1 homolog 2, mismatch repair system component pseudogene 6 [Homo sapiens]	NC_000007.14:73093644-73096950	No
*PMS2P7*	PMS1 homolog 2, mismatch repair system component pseudogene 7 [Homo sapiens]	NC_000007.14:c73006080-73016375	Yes
*PMS2P8*	PMS1 homolog 2, mismatch repair system component pseudogene 8 [Homo sapiens]	NC_000007.14:73037373-73040804	No
*PMS2P9*	PMS1 homolog 2, mismatch repair system component pseudogene 9 [Homo sapiens]	NC_000007.14:77039480-77053038	No
*PMS2P10*	PMS1 homolog 2, mismatch repair system component pseudogene 10 [Homo sapiens]	NC_000007.14:c75327776-75324481	No
*PMS2P11*	PMS1 homolog 2, mismatch repair system component pseudogene 11 [Homo sapiens]	NC_000007.14:77011447-77025554	No
*PMS2P12*	PMS1 homolog 2, mismatch repair system component pseudogene 12 [Homo sapiens]	NC_000007.14:102337315-102339139	No
*PMS2CL*	PMS2 C-terminal like pseudogene [Homo sapiens]	NC_000007.14:c6735305-6751601	No
*LOC441259*	PMS1 homolog 2, mismatch repair system component pseudogene [Homo sapiens]	NC_000007.14:c75299806-75296553	No

## Data Availability

The original contributions presented in the study are included in the article; further inquiries can be directed to the corresponding author.

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
