# Peer review of "A Pathogenic Variant Reclassified to the Pseudogene PMS2P1 in a Patient with Suspected Hereditary Cancer"

_ijms, 2023, doi:10.3390/ijms24021398_

Round 1

Reviewer 1 Report

The paper entitled with “A pathogenic variant reclassified to the pseudogene PMS2P1 in a patient with suspected hereditary cancer” reports a novel pathogenic PMS2 variant which is related with breast cancer in one female patient. In general, this paper is informative. The paper includes a nice introduction about the LS but I would like to see a more detailed description about the PMS2 gene since that is the one that this paper focused on.  Since they found the frequency of 100% in terms of this variant in the analyzed individual, relatives, and unrelated individuals in this study, the correlation between this PSM2 pseudogene and the cancer reported in this paper was weak. Therefore, I suggest the author not to use definitive statements unless they can provide more evidence. Also, the figure legend of figure 1 is confusing and it doesn’t describe the figure properly. Besides, there are typos and some grammatical mistakes in the paper. Please fix them.

Reviewer 2 Report

In this paper, the authors reported the case of a 44-year-old woman diagnosed with breast cancer in which was found a pathogenic variant in PMS2 gene, determiing that the variant is associated with the PMS2P1 pseudogene. The manuscript is quite interesting, however it needs some modifications.

- In introduction the authors should indicate that there are specific therapies for the carriers of mutations in MMR or in other genes, since they then talk about it later;
- It would be more correct to specify the genes analyzed with NGS;
- In the text reference is made to figures 1, 2 and 3, but they do not appear in the manuscript;
- The authors should specify whether variants have also been identified in other genes;
- It would be preferable that in the text and tables the identified variant was always reported with both names (c.1A>T, p.Met1Leu).
